# Tsuchime-like Aluminum Film to Enhance Absorption in Ultra-Thin Photovoltaic Cells

**DOI:** 10.3390/nano13192650

**Published:** 2023-09-26

**Authors:** Mikita Marus, Yauhen Mukha, Him-Ting Wong, Tak-Lam Chan, Aliaksandr Smirnov, Aliaksandr Hubarevich, Haibo Hu

**Affiliations:** 1Centre for Advances in Reliability and Safety (CAiRS), Unit 1212–1213, 12/F, Building 19W, Hong Kong Science Park, Pak Shek Kok, New Territories, Hong Kong, China; mikita.marus@cairs.hk (M.M.); abel.wong@cairs.hk (H.-T.W.); tom.chan@cairs.hk (T.-L.C.); 2Laboratory for Information Display and Processing Units, Belarusian State University of Informatics and Radioelectronics, 6 P. Brovki, 220013 Minsk, Belarus; muha@bsuir.by (Y.M.); smirnov@bsuir.by (A.S.); 3Department of Electrical and Electronic Engineering, Hong Kong Polytechnic University, Hong Kong, China

**Keywords:** thin film, nanostructure, solar energy harvesting, solar cells, nanomaterial

## Abstract

Ultra-thin solar cells enable materials to be saved, reduce deposition time, and promote carrier collection from materials with short diffusion lengths. However, light absorption efficiency in ultra-thin solar panels remains a limiting factor. Most methods to increase light absorption in ultra-thin solar cells are either technically challenging or costly, given the thinness of the functional layers involved. We propose a cost-efficient and lithography-free solution to enhance light absorption in ultra-thin solar cells—a Tsuchime-like self-forming nanocrater (T-NC) aluminum (Al) film. T-NC Al film can be produced by the electrochemical anodization of Al, followed by etching the nanoporous alumina. Theoretical studies show that T-NC film can increase the average absorbance by 80.3%, depending on the active layer’s thickness. The wavelength range of increased absorption varies with the active layer thickness, with the peak of absolute absorbance increase moving from 620 nm to 950 nm as the active layer thickness increases from 500 nm to 10 µm. We have also shown that the absorbance increase is retained regardless of the active layer material. Therefore, T-NC Al film significantly boosts absorbance in ultra-thin solar cells without requiring expensive lithography, and regardless of the active layer material.

## 1. Introduction

Decarbonization is a major challenge for the global society [1,2,3]. While we have significantly reduced energy consumption in household and industrial electrical devices over the past decade, the electricity demand continues to grow due to the increasing use of electric vehicles, artificial intelligence, and visual information devices [4,5,6,7]. Additionally, the lifespan of handheld devices is decreasing, and augmented and virtual reality devices are becoming more prevalent [8,9]. To address these challenges, we must focus on clean and sustainable energy sources, such as solar power [10].

Ultra-thin photovoltaic cells (PVs) offer strong advantages such as saving materials, reducing the deposition time, and providing the possibility of using absorber materials with short carrier diffusion lengths [11,12]. However, the efficiency of thin and especially ultra-thin PV cells is limited by weak light absorption caused by low absorption coefficients and reduced active layer thickness [13,14,15]. To improve absorption, it is necessary to minimize reflection on the surface of the solar panel and increase the optical path length [16,17]. This can be achieved through micro- and nanoscale photonic structures [18], but these typically require expensive top–down processes such as lithography [19,20]. Another approach relies on increasing the photon absorption mediated by local plasmons. This could be cheap and relatively easy to apply but is restricted to particular materials and affects a limited wavelength range [21,22].

Here, we show that a self-forming nanostructure—Tsuchime-like nanocrater aluminum (T-NC Al) film—greatly enhances absorption in an ultra-thin solar cell. At the same time, it eliminates the need for lithography and other costly manufacturing processes and becomes more effective as the cell gets thinner. The nanostructure resembling a nanoscale Tsuchime (or hammertone) patterned surface results from the electrochemical nanoporous (NP) anodization of Al followed by etching away the NP alumina (Al_2_O_3_) layer [23]. This method is cost-effective, lithography-free, easily scalable, and allows for the diameter and depth of resulting nanocraters (NCs) to be tuned in a wide range [23,24,25,26]. Moreover, it allows for the production of ordered micro/submicro- and nanopatterned surfaces by iterating the procedure in several steps of anodization followed by etching [27,28]. In addition, metallic micro- and nanostructures fully comply with flexible optoelectronic devices, including flexible solar cells and displays [5,29,30,31].

We conducted a theoretical study of Tsuchime-like NC (T-NC) film as a functional layer of a structure consisting of the absorbing layer covering T-NC and resembling a simplified ultra-thin solar cell. We found the optimum diameter of NCs in the nanopatterned film for different active layer thicknesses to maximize the optical path in the structure. The effect of using a T-NC nanostructure is adversely related to the thickness of the active layer; the effect of T-NC is stronger when the active layer is thinner and weakens with the increase in the thickness of the active layer of the solar cell. When applied in PV cells with active layer thicknesses of 500 nm, 1 µm, 3 µm, and 10 µm, T-NC film increased the average absorbance (Δλ = 400–1100 nm) by 80.3%, 62.8%, 41.5%, and 20.0%, respectively.

Added to the above, the wavelength range of increased absorption depends on Si thickness. The peak of absolute absorbance increase moves from 620 nm to 950 nm as the thickness of Si increases from 500 nm to 10 µm. So, thinner PV cells benefit in the visible spectrum, while thicker cells benefit in the near-infrared range. Finally, the increase in absorbance remains regardless of the active layer material. In the last part of the study, we changed the active layer material with a hybrid organic–inorganic formamidinium lead iodide (FAPbI_3_) and organic blend of poly[[4,8-bis[(2-ethylhexyl)oxy]benzo[1,2-b:4,5-b′]dithiophene-2,6-diyl][3-fluoro-2-[(2-ethylhexyl)carbonyl]thieno[3,4-b]thiophenediyl]]: [6,6]-phenyl C71-butyric acid methyl ester (PT7B:PC71BM). In both cases, T-NC increased the average absorbance, and the effect was stronger for thinner active layers. For 300 nm thick solar cells, T-NC increased the average absorbance in FAPbI3- and PT7B:PC71BM-based cells by 45.8 and 34.9%, respectively. Therefore, T-NC Al film significantly increases the efficiency of ultra-thin solar cells without using expensive lithography, regardless of the active layer material.

## 2. Materials and Methods

Figure 1 schematically explains the concept behind increasing absorption in thin and ultra-thin Si solar cells using a T-NC Al film. When light interacts with the planar Al surface (Figure 1a,b), there is a single process of reflection of light. The effective optical path of the reflected light (*L_R_*) is equal to the optical path of the transmitted light through the active layer (*L_T_*). In the case when light collides with the T-NC film (Figure 1c,d), it reflects in directions other than in the case of a planar surface. This includes multiple reflections from the inner side of the NC. As a result of multi-reflection, the effective optical path inside the Si active extends (*L_R_*_1_ + *L_R_*_2_ + … > *L_T_*), leading to increased absorption in the cell. Figure 1e shows a scanning electron microscope (SEM) image of the experimentally obtained T-NC Al surface with a diameter of NC *d_NC_* = 300 nm. As can be seen from the figure, the nanopatterned structure resembles a crater left by a hammer on a metallic surface—the so-called hammertone or Tsuchime technique [32]. Experimental details of the electrochemical NP anodization and etching of Al film are fully described in our previous work (Ref. [17]). Figure 1f shows the lateral SEM image of the same T-NC Al film.

The unit cell of whole solar cell structures, including the Si active layer and Al planar electrode and electrode with a Tsuchime-like nanostructured surface, was transferred into the commercial COMSOL simulator (Figure 2a,b) [33]. The incident light source from 400 nm to 1.1 µm was illuminated along the *Z*-axis and placed above the Si layer. The transmittance and reflectance monitors were accordingly located below the Al layer and above the light source. The periodic boundary conditions and perfectly matched layers (PML) were applied perpendicular and parallel to the *Z*-axis. Figure 2c shows a 3D view of a single NC unit of T-NC Al film, where D is the diameter of NC. Optical constants of Si and Al were taken from Refs. [34,35]. It should be noted that the COMSOL simulator has limitations in cases where the nanostructure under consideration for a solar cell falls within the range of influence of plasmonic effects for a given nanostructure material [36].

Numerical optical simulations using the finite element method or finite difference time domain include the coherent propagation of light. This may result in wave interference fringes when simulating multilayer structures with an effective thickness of one or more layers equal to or greater than the incoming wavelength. Such wave interference fringes are not observed experimentally in most cases because structure layers are not so perfectly planar. To avoid this issue, some scientists applied phase matching and phase elimination approaches [37]. Unfortunately, these approaches do not work for all cases. Here, we propose a seasonal autoregressive integrated moving average (SARIMA)-reinforced zero-phase filter to artificially smooth the undesired fringes of the absorbance spectra [38,39]. In signal processing, high-frequency noise can be reduced through a zero-phase filter, keeping the phase of low-frequency signals unchanged [38,39,40,41,42]. However, a zero-phase filter is non-casual and relies on the data in future; hence, in our case, the last few data points could not be obtained. To extend the data domain, we employed SARIMA with the parametric notation of
(1)$$ARIMA(p,d,q)\times \left(P,D,Q\right)S,$$
where *p* is non-seasonal AR order; *d* is non-seasonal differencing; *q* is non-seasonal MA order; *P*, *D*, and *Q* are their seasonal counterparts; and *S* is the seasonal period. Formally, the model could be written as
(2)$$\Phi \left(B^{S}\right)\phi \left(B\right)\left(x_{t}-\mu \right)=\Theta \left(B^{S}\right)\theta \left(B\right)\omega_{t},$$
where *B* is the backward shift operator, *ϕ*(*B*) = 1 − *ϕ*_1_*B* − ⋯ − *ϕpBp* is the non-seasonal AR operator, *θ*(*B*) = 1 + *θ*_1_*B* + ⋯ + *θqBq* is the non-seasonal MA operator, *Φ*(*B*) = 1 − *Φ*_1_*B^S^* − ⋯ − *Φ_p_B^PS^* is the seasonal AR operator, *Θ*(*B*) = 1 + *Θ*_1_*B^S^* + ⋯ + *Θ_q_B^QS^* is the seasonal MA operator, and *ω_t_* is the error term. Then, we can use the iterative method to find the best seasonal ARIMA parameter and choose a zero-phase filter with suitable order to smooth out the fringes of the absorbance spectra. Figure 3 shows an example of applying a SARIMA-reinforced zero-phase filter on simulated absorbance spectra of PV cells with a planar Al (Figure 3a) and T-NC Al layer with *d_NC_* = 800 nm (Figure 3b) and a 3000 nm thick active layer.

## 3. Results and Discussion

### 3.1. Diameter of Nanocraters (d_NC_)

In the first phase of the study, we estimated the influence of the diameter of NCs (*d_NC_*) on the level of light absorption at a fixed thickness of the Si active layer. Figure 4 shows the optical absorbance of simulated Si PV cells with planar Al and T-NC substrates. The diameter of NCs (*d_NC_*) varied from 0 nm (planar Al) to 800 nm, as such diameters can be easily obtained during Al anodization [43]. Starting from *d_NC_* = 200 nm, T-NC gained a significant advantage over the planar Al substrate under Si and reached a maximum increase of 62.8% at *d_NC_* = 800 nm. When *d_NC_* was compatible with an incoming wavelength, the light got reflected repeatedly from the inner side of the NCs and thus increased the reflective optical path and absorbance in the Si layer. The effect of the T-NC film increased when the thickness of the active layer decreased. For example, the absorbance of a PV cell having a 1000 nm thick Si active layer increased by 20% at λ = 590 nm (from 51.1 to 61.5%), while for PV cells having *H_Si_* = 3000 nm, it increased by only 10.4%, while for a 10 µm thick active layer, it remained the same regardless of T-NC.

It is remarkable that the use of T-NC Al film effectively eliminated the need for a thicker active layer. While the difference between the absorbance in PV cells with 500, 1000, and 3000 nm thick Si active layers was significant when the Al substrate was planar, it was virtually eliminated with the T-NC Al film where *d_NC_* = 800 nm. Indeed, the average absorbance of PV cells with 500, 1000, and 3000 nm thick Si active layers jumped from 33.5, 37.2, and 44.3%, respectively, to over 60% when *d_NC_* = 800 nm (at wavelength range Δλ = 400–1100 nm). For a PV cell with a 500 nm thick Si active layer, this translated into an over 80.3% boost over the cell with a planar Al substrate. T-NC-enhanced PV cells with 1000 and 3000 nm Si active layers gained 64.3 and 41.8% relative increases in the average absorbance over solar cells with planar Al. However, the PV cell with a 3000 nm thick Si active layer and *d_NC_* = 800 nm reached the highest average optical absorption in the given wavelength range (>62.8% at Δλ = 400–1100 nm).

### 3.2. Thickness of the Active Silicon Layer (H_Si_)

During the second phase of the study, we determined the optimal thickness of the active silicon layer to provide synergy with the T-NC Al film. Figure 5 shows the optical absorbance of the simulated ultra-thin PV cells with planar Al (Figure 5a) and T-NC substrates (Figure 5b) with varied thicknesses of the Si active layer (*h_Si_* = 500 nm to 10 µm) and fixed *d_NC_* = 800 nm. As shown in Figure 5a, planar structure strongly depends on the thickness of the active layer. The planar PV cell with a 10 µm thick active layer outperformed the PV cell with a 500 nm thick active layer by over 78.5%. However, with T-NC film, the difference between the 500 nm and 10 µm thick active layer was nearly eliminated. The average absorbance in PV cells with T-NC film (dNC = 800 nm) with 500 nm and 10 µm thick active layers was 60.3% and 60.8%, respectively. Table 1 summarizes the increase in average absorption due to the use of a T-NC structure with d_NC_ = 800 nm as a function of the active layer thickness. The effect of the T-NC structure was four times stronger for a PV cell with an active layer thickness of 500 nm compared to a PV cell with an active layer thickness of 10 microns.

Added to the forgoing, the absorption range is directly dependent on the thickness of the active layer. Figure 5c shows the absolute difference in the total optical absorbance of simulated thin and ultra-thin Si PV cells with planar Al- and T-NC-modified substrates with varied thicknesses of the Si active layer (*H_Si_* = 500 nm to 10 µm) and *d_NC_* = 800 nm. As shown in Figure 5d, the peak absolute difference in absorbance was observed at λ = 950 nm: there was an about 42% gain over the structure with a flat aluminum layer, regardless of the thickness of the active silicon layer. Interestingly, the peak absorption difference in the active element (shown in Figure 5d) migrated from the visible to the near-infrared range as the active layer thickness increased. The maximum absorbance difference inside the Si active element with a thickness of 500 nm was at λ = 620 nm. On the other hand, for structures where H_Si_ = 1, 3, and 10 microns, it was at 680, 850, and 950 nm, respectively. The absolute increase in the average absorbance of T-NC with 800 nm NCs over planar Al was 27.3, 23.2, 17.8, and 2.6% for H_Si_ = 500 nm, 1 µm, 3 µm, and 10 µm, respectively.

### 3.3. Effect of the Active Layer Material

To investigate the effect of the active layer material on the efficiency of the T-NC nanostructure, we changed the inorganic silicon active layer with (a) a hybrid organic–inorganic active layer based on formamidinium lead iodide (FAPbI3); and (b) an organic active layer based on poly[[4,8-bis[(2-ethylhexyl)oxy]benzo[1,2-b:4,5-b′]dithiophene-2,6-diyl][3-fluoro-2-[(2-ethylhexyl)carbonyl]thieno[3,4-b]thiophenediyl]]: [6,6]-phenyl C71-butyric acid methyl ester (PT7B:PC71BM). Perovskite solar cells (PVCs) are promising candidates for the next generation of photovoltaics due to their outstanding photoelectric properties and low-cost processing techniques [44,45,46]. Generally, the thickness of the FAPbI_3_ perovskite active layer varies in the range of 300 nm to 1 µm across the reported studies [44,45,46,47,48]. Figure 6a shows the optical absorbance of a FAPbI_3_ perovskite PV cell with the corresponding thicknesses of the active layer.

As in the case of the silicon active layer, the absorption in the cells with the FAPbI_3_ perovskite active layer increased for both material thicknesses (shown in Figure 6a), and the effect was stronger for thinner perovskite layers. For 300 nm and 1000 nm thick FAPbI_3_ layers, the average absorbance increased by 45.8 and 37.5%, respectively. Similar to the case with silicon active layers, the difference in average absorbance between the PV cells with 300 nm and 1000 nm thick active layers dropped from 11.0 to only 4.6%. It is important to mention that the sole enhancement of the light absorption is not the only mechanism to increase the efficiency of perovskite solar cells where the large increase in the final efficiency of a cell has been observed due to the plasmon effect related to the influence of metallic components on the internal electricity of a cell [49,50,51]. However, the current study focuses on Al nanostructures, where the localized surface plasmon resonance (LSPR) affects the ultraviolet (UV) wavelength region below 400 nm [24,52]. Covering the T-NC nanostructure with an ultra-thin (5–10 nm) layer of another metal such as gold (Au) or silver (Ag) onto the T-NC nanostructure can shift the plasmonic effect from the UV towards the visible wavelength range [49].

Figure 6b shows the optical absorbance of a PT7B:PC71BM organic PV cell with the same 300 nm and 1000 nm thicknesses of the active layer. Solution-processable blends such as compounds of PT7B and PC71BM offer high efficiency, excellent flexibility, and low fabrication costs [53,54,55,56]. The choice of the thickness of the organic active layer was dictated primarily by the fact that we wanted to compare the effect of different materials on the efficiency of the nanostructure. Thicknesses greater than 300 nm are practically insensible due to the short diffusion length of an exciton in the given organic compound [54,55,57]. As can be seen in Figure 6b, increasing the thickness of the PT7B:PC71BM active layer resulted in only a marginal increase in the average absorbance from 51.4 to 52.7% without a T-NC nanostructure. With the inclusion of a T-NC nanostructure, the average absorbance increased from 67.4 to 69.7%. It follows that the absorbance increased by 34.1 and 32.3% for 300 nm and 1000 nm thick active layers, respectively. The results indicate that a thinner PT7B:PC71BM active layer offers the same high optical absorbance coupled with the T-NC nanostructure. Moreover, as we mentioned before, a thin PT7B:PC71BM layer renders more efficient devices due to the short diffusion length of the exciton. Table 2 summarizes the increase in the average absorbance in FAPbI3- and PTB7:PC71BM-based PV cells owing to the T-NC structure with a d_NC_ = 800 nm.

It should be noted that the implementation of tight adjoining between the active layer material and the T-NC Al nanostructure will strongly influence both the optical gain and the electrical gain of the solar cell. Roughness induced by the nanostructured Al electrode would deteriorate the contact characteristics of the active layer and the Al electrode, which might worsen the transport and collection of charge carriers at the contact interface and thus weaken the performance of solar cells [58,59,60]. However, in the case of perfect adjoining between the active layer material and the T-NC Al nanostructure, the latter would facilitate the collection of charge carriers and improve the short-circuit current density (J_SC_) [59].

## 4. Conclusions

In this contribution, we propose a T-NC nanostructured aluminum surface for improving absorption in ultra-thin solar cells, which can be obtained without lithography—in the cost-effective process of nanoporous anodization followed by oxide etching. For ultra-thin silicon solar cells with an active layer thickness of 500 nm, a T-NC Al structure theoretically increases the average absorption by over 80.3%. Moreover, the wavelength range where the absorption increases the most varies depending on the thickness of the active layer. In the case of a solar cell with an active layer thickness of 500 nm, the absorbance increase maximum is at λ = 620 nm, and it moves to λ = 950 nm for a solar cell with an active layer thickness of 10 µm. A T-NC aluminum layer eliminates the need for a thick active layer; solar cells with an active layer thickness of 500 nm and 10 microns show comparable results—60.3 and 60.9% average absorption—with the same NC diameter d_NC_ = 800 nm. Finally, the effect of T-NC nanostructure is retained regardless of the active layer material. Solar cells based on a hybrid organic–inorganic perovskite FAPbI_3_ layer and organic blend PTB7:PC71BM with the active layer thickness of 300 nm improved the average absorbance by 45.8 and 34.1%, respectively, with a T-NC structure with the same d_NC_ = 800 nm. A further direction of research may include employing a plasmonic effect, for example, by adding an ultra-thin layer of another metal such as Au or Ag onto the T-NC nanostructure.

## Figures and Tables

**Figure 1 nanomaterials-13-02650-f001:**
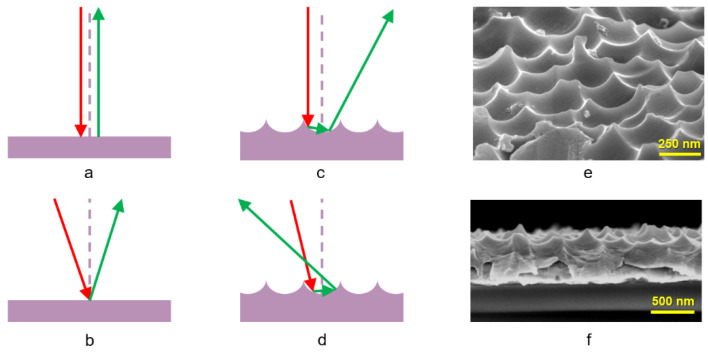
(**a**–**d**) Schematic representation of increasing absorption in ultra-thin Si solar cells using a T-NC Al film. The effect results from elongating the optical path inside the active silicon layer. Here, (**a**,**b**) represent cases when light reflects from a plain Al surface [*L_R_* ~ *L_T_*]; (**c**,**d**) represent cases when light faces multiple reflections from a T-NC Al film surface, resulting in a longer optical path [*L_R_*_1_ + *L_R_*_2_ + … > *L_T_*]. The red and green lines represent incident and reflected light, respectively. (**e**) SEM image of the T-NC Al film fabricated by our group [23]. Scale bar: 250 nm. (**f**) Lateral SEM image of same T-NC Al film. Scale bar: 500 nm. The manufacturing process is fully described in [17].

**Figure 2 nanomaterials-13-02650-f002:**
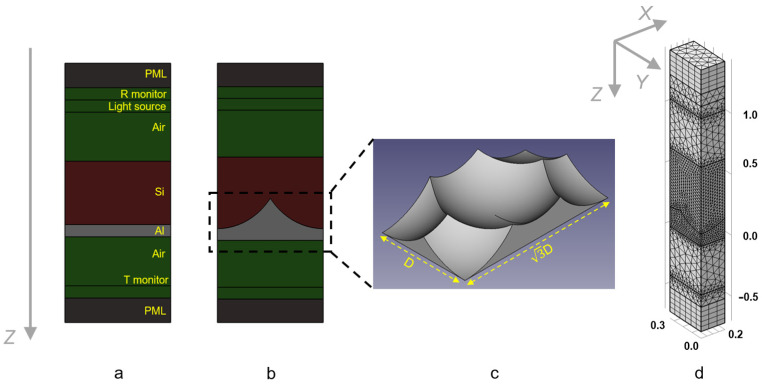
(**a**) The unit cell of the simulated PV element, including the Si active layer and planar Al film, surrounded by air. A light source is placed above the structure’s layer, while transmittance and reflectance monitors are located below the Al layer and above the light source. The periodic boundary conditions and PML were applied perpendicular and parallel to the *Z*-axis. (**b**) The unit cell of the simulated PV element, including the Si active layer and T-NC-treated Al film, is surrounded by air. (**c**) Three-dimensional view of a single NC unit of T-NC Al film. Here, *D* is the diameter of NC. (**d**) Examples of mesh distribution inside the unit cell. The unit cell was set to d/2 × 3^1/2^d/2 = 3^1/2^d/4 to accelerate the calculation. The grid varied from 1 nm (interfaces and monitors) to 25 nm (Si) and 20 nm (Al). Dimensions marked in the figure are in µm.

**Figure 3 nanomaterials-13-02650-f003:**
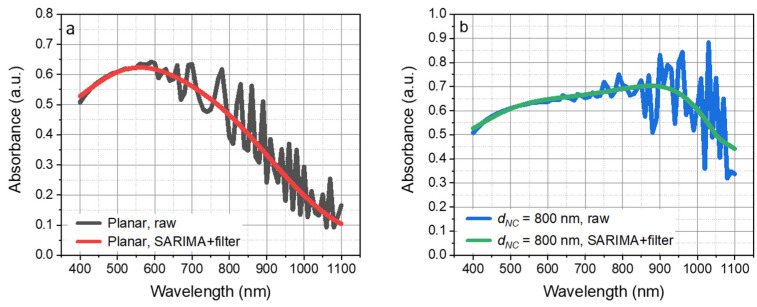
Example of employing SARIMA + filter on coherent simulated absorbance spectra of PV cell with 3000 nm thick active layer: (**a**) planar Al substrate; (**b**) T-NC Al substrate with *d_NC_* = 800 nm.

**Figure 4 nanomaterials-13-02650-f004:**
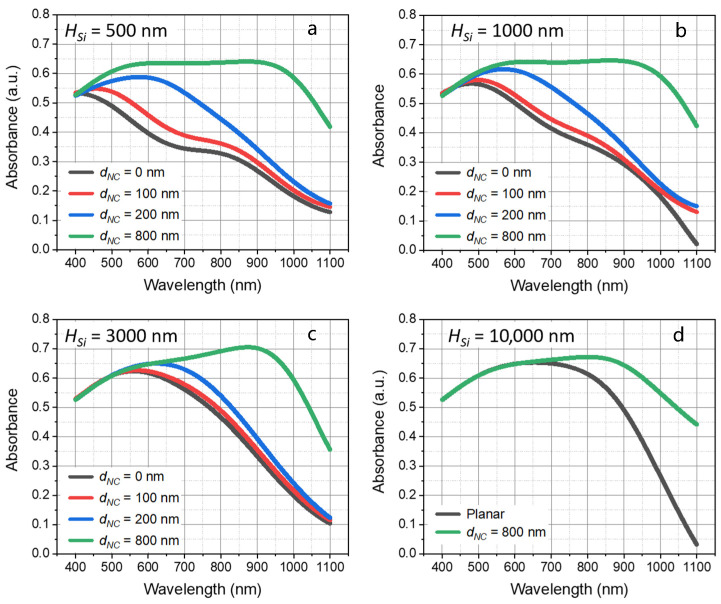
Optical absorbance of thin and ultra-thin Si PV cells with planar Al and T-NC Al substrates with varied diameters of the NCs (*d_NC_* = 0 nm to 800 nm). The thickness of the Si active layer (*H_Si_*) equals (**a**) 500 nm, (**b**) 1000 nm, (**c**) 3000 nm, and (**d**) 10 µm. The insets show the diameters of the NCs (*d_NC_*) ranging from 0 nm (planar Al) to 800 nm.

**Figure 5 nanomaterials-13-02650-f005:**
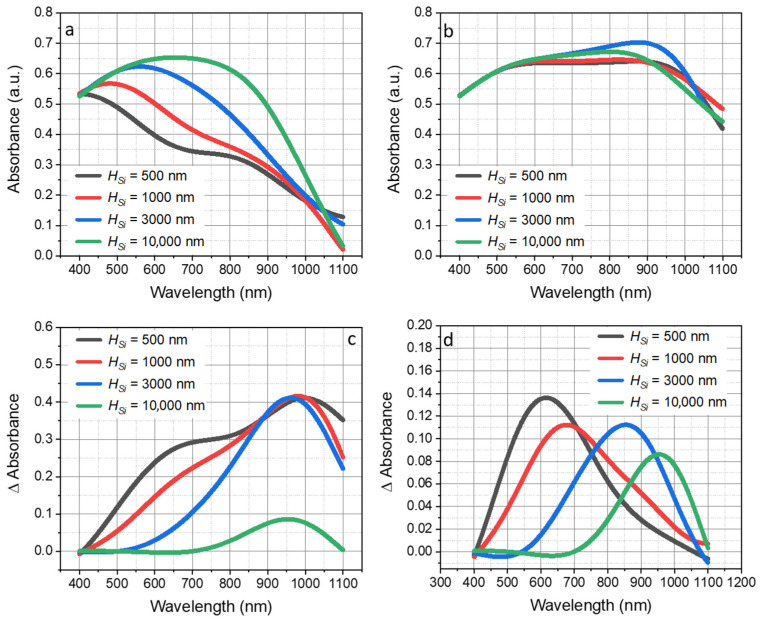
(**a**,**b**) Optical absorbance of ultra-thin Si PV cells with planar Al (**a**) and T-NC Al substrates (**b**) with varied thickness of Si active layer (*H_Si_* = 500 nm to 1 µm). The insets show the thickness of the Si active layer (*H_Si_*): ranging from 500 nm to 10 µm. (**c**,**d**) Absolute difference in total absorbance (**c**) and absolute absorbance difference in the active layer (**d**) of simulated ultra-thin PV cells with planar Al and T-NC Al substrates with varied thicknesses of Si active layer (*H_Si_* = 500 nm to 10 µm) and *d_NC_* = 800 nm.

**Figure 6 nanomaterials-13-02650-f006:**
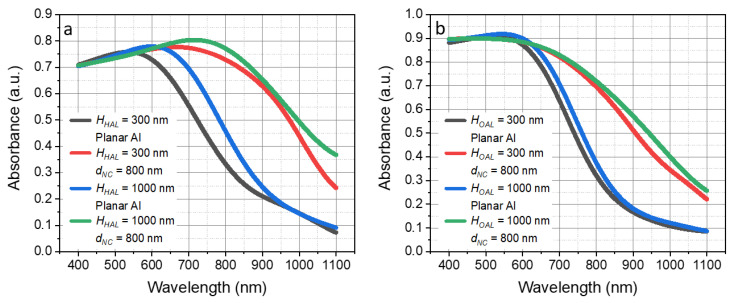
Optical absorbance of thin and ultra-thin formamidinium lead iodide (FAPbI3) perovskite PV cell (**a**), and poly[[4,8-bis[(2-ethylhexyl)oxy]benzo[1,2-b:4,5-b′]dithiophene-2,6-diyl][3-fluoro-2-[(2-ethylhexyl)carbonyl]thieno[3,4-b]thiophenediyl]] (PTB7):[6,6]-phenyl C71-butyric acid methyl ester (PC71BM) organic PV cell (**b**). Thickness of the active layer varied from 300 to 1000 nm, accordingly.

**Table 1 nanomaterials-13-02650-t001:** An increase in the average absorption in PV cells owing to the T-NC structure with a *d_NC_* of 800 nm, depending on the thickness of the active layer.

	0.5 µm	1.0 µm	3.0 µm	10 µm
AT−NCaveAPlanarave, %	80.3	62.8	41.5	20.0

**Table 2 nanomaterials-13-02650-t002:** An increase in the average absorption in FAPbI_3_- and PTB7:PC71BM-based PV cells owing to the T-NC structure with a d_NC_ of 800 nm, depending on the thickness of the active layer (H, µm).

	0.3 µm	1.0 µm
AT−NCaveAPlanarave( FAPbI_3_), %	45.8	37.5
AT−NCaveAPlanarave( PTB7:PC71BM), %	34.1	32.3

## Data Availability

Data underlying the results presented in this paper are available from the corresponding authors upon reasonable request.

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
