# Peer review of "Tsuchime-like Aluminum Film to Enhance Absorption in Ultra-Thin Photovoltaic Cells"

_nanomaterials, 2023, doi:10.3390/nano13192650_

Round 1
Reviewer 1 Report (New Reviewer)
This is a simulation job. There is no comparison with experimental results, which translates into lower quality of work.
The word PV (photovoltaic) appears several times in the manuscript, but there are no measured cells. There are only optical measurements of thin layers, which in this case correspond to materials that can be used in solar cells.
Even the simulation of optical properties is carried out in structures that do not correspond to the structure of a solar cell. A solar cell is made up of selective electron and hole layers, anti-reflection layers (in the case of crystalline silicon), Transparent Conductive layers (ITO, FTO,...) in the case of organic and perovskite solar cells, etc.
The work has little or no relationship to solar cells.
The inclusion of the word photovoltaic cells in the title leads to confusion.
This fact together with the fact that there is no comparison with experimental results is, in my opinion, the negative aspects of the manuscript, and in my case it leaves me in doubt as to whether it should be published.
The work is well done. I leave it to the editor's discretion whether the work is suitable for publication or not.
Author Response
We would like to thank Reviewer I for evaluating our work and for the valuable commentary. Indeed, this is a theoretical investigation of employing a real nanostructure (T-NC nanostructure resulting from electrochemical anodizing of aluminum and etching off the alumina) for enhancing the absorbance in ultra-thin cells consisting of absorbing layer (inorganic, hybrid organic-inorganic, and organic compound) and metallic (aluminum, Al) nanostructure beneath. Such T-NC nanostructure has a broad range of potential use cases, not only solar cells, – basically any application, which benefits from increasing the optical pathway inside the structure would benefit from T-NC nanostructure.
However, photovoltaics and ultra-thin solar cells are of particular interest here, since these types of optoelectronic devices (a) require maximizing the light path inside the active layer, and (b) require efficient charge transfer from the active layer. These two points present a trade-off since, on the one hand, you need the active layer as thick as possible to maximize the light path, but, on the other hand, you need the active layer thin enough to efficiently extract the charge. From this perspective, the T-NC nanostructure addresses this trade-off by increasing the optical path in the “active” layer with the help of the nanostructure beneath and without the need to increase the thickness of the “active” layer.
Summary of the reasons why the T-NC structure is of interest, particularly for solar cells:
- T-NC nanostructure omits lithography or any other costly/vacuum processes. It can be formed on any scale through a simple method of anodizing aluminum and on various substrates, including flexible ones [1].
- T-NC increases the absorbance in a structure formed of the “active” layer (inorganic, organic, or hybrid) and T-NC nanostructure beneath up to 80%, depending on the “active” layer thickness.
- T-NC works regardless of the material above – data added to the manuscript for the hybrid material (FAPbI3 perovskite) and organic compound (PTB7:PC71BM).
- Absorbance increase maxima depends on the thickness of the layer above and can be controlled accordingly.
- T-NC nanostructure efficiency is adversely related to the thickness of the layer above, which is beneficial from the perspective of the trade-off between absorption and charge transfer.
Considering the above, even though we currently lack the capacity to manufacture the cell, we have shown accordingly: (a) that the nanostructure can be made via a known method; (b) T-NC significantly increases the absorbance in the ultra-thin structures composed of the “active” layer over T-NC (based on theoretical investigation); (c) T-NC compliments the existing trade-off in ultra-thin solar cells between the absorption and charge transfer (see below). The finite element method (FEM) and the finite-difference time-domain (FDTD) methods are proven approaches to accurately analysing micro- and nanostructures featured in thousands of research articles including the Nanomaterials journal [2]. For example, J. Wei et al. employed the latter method in the numerical study of complementary nanostructures for light trapping in CQD solar cells. Here we employ the FEM method to study the nanostructure for light trapping in ultra-thin solar cells, which has not been considered from this perspective before. In J. Wei’s article, they demonstrate that the nanostructure can be made through the method of nanosphere lithography (NSL) and theoretically increases the absorbance in the CQD solar cell. In the current article, we demonstrate that the T-NC nanostructure can be made through the simple method of electrochemical anodization and theoretically increases the absorbance in the ultra-thin solar cell (regardless of active layer material). We understand the Reviewer’s concern, nevertheless, this hybrid method is widely employed in the practice of nanostructure research, including in the Nanomaterials journal.
Moreover, the study on various overcoating materials (organic and hybrid in addition to inorganic active layer) supported our initial theoretical study with silicon over T-NC (see Figure 6 in the paper). The average absorbance of the structure consisting of a 300 nm thick perovskite “active” layer and T-NC with dNC = 800 nm beneath increased 1.46 times, while for 1000 nm thick perovskite active layer it increased 1.37 times. In the case of the structure consisting of PTB7:PC71BM and the same T-NC with dNC = 800 nm beneath, T-NC increased the optical absorbance by 1.35 and 1.33 times for 300 nm and 1000 nm thick organic layer, accordingly. Here, the organic active layer can be as thin as possible to complement charge transfer/extraction properties since short exciton diffusion length will hinder the device's efficiency [3].
A solar cell is made up of selective electron and hole layers, anti-reflection layers (in the case of crystalline silicon), Transparent Conductive layers (ITO, FTO,...) in the case of organic and perovskite solar cells, etc.
We want to thank Reviewer I for the insight. Indeed, most of the current solar cells consist of multiple layers including electron transport layers, hole transport layer, absorbing active layer between them, electrodes, and anti-reflection layers. However, the purpose of the paper was to demonstrate the effect of increasing absorption with the help of T-NC nanostructure by increasing the optical pathway in the coating (“active”) material. Since there are a variety of “active” materials used for photovoltaics and a multitude of possible configurations (including single-layer solar cells, double-layer solar cells, “tandem” solar cells, and many other types), we chose the simplest structure of a single-layer coated over T-NC nanostructured “electrode”, which is sufficient to demonstrate the concept of T-NC beneath the “active” layer without limiting to particular structures and materials. As can be seen in the paper, perovskite was not the only material participating in the study, which included inorganic silicon and organic polymer alternatives. Each of these materials benefits from a different configuration of the solar cell; however, we demonstrated that each of these materials might benefit from using a T-NC nanostructure, regardless of the origin of the material because the effect of T-NC lies in increasing optical pathway in the coating material. Using simplified photovoltaic cell structures for studying solar energy concepts is a normal practice within the literature [4]. Moreover, we changed the introduction part presenting a structure in lines 59-61. It now says, “We conducted a theoretical study of Tsuchime-like NC (T-NC) film as a functional layer of a structure consisting of the absorbing layer covering T-NC resembling a simplified ultra-thin solar cell and found the optimum diameter of NCs in the nanopatterned film for different active layer thicknesses to maximize the optical path in the solar cell.” to explain the reader that this is a simplified structure resembling an ultra-thin solar cell.

Reviewer 2 Report (New Reviewer)
The authors report on the simulation study of improving/enhancing absorption for solar cell applications using their proposed “Tsuchime-like aluminum film”. They show the absorption change with texture diameters and active layer thicknesses to verify their concept. The manuscript is well written, but the reviewer has some serious questions. Many research works have been done to improve the optical absorption for solar cell applications but there is a trade-off between optical gain and electrical gain. The higher textured surface is not always good for the device performance. So, the authors need to mention this point and if possible, the authors show some data on this matter.
Author Response
We would like to thank Reviewer II for the evaluation of our work and for the valuable comment. Reviewer II indeed raised an important point about the existing trade-off between efficiency of absorption and efficiency of charge extracting. On the one hand, roughness induced by the nanostructured Al electrode would deteriorate the contact characteristics of the active layer and the Al electrode, which might worsen the transport and collection of charge carriers at the contact interface and thus weaken the performance of solar cells [5,6]. On the other hand, the interpenetrated structure of the active layer and Al electrode provides relatively shorter routes for electrons to travel to the cathode, and the enlarged active layer/electrode contact interface facilitates charge carriers collection, which, in turn, improves the short-circuit current density (JSC) [5]. For example, X. Li et al. proposed an interpenetrated (micro-nanostructured) to promote light absorption in polymer solar cells, which also promoted charge collection efficiency and JSC [5]. Planar device reached JSC = 6.93 mA/cm2, while the device with micro-nanostructured Al electrode reached JSC = 8.17 mA/cm2 due to interpenetrated Al electrode with the polymer active layer slightly expanding the contact active area. X. Li et al. also mention that the open circuit voltage (VOC) and fill factor (FF) remained intact for the micro-nanostructured Al electrode. Considering the above, we might expect similar behaviour from the T-NC nanostructured Al when considering the trade-off between optical gain and electrical gain. We must note here that the quality of the contact between the active layer and the T-NC would affect not only electrical gain, but the optical gain as well. Introducing the gap between the active layer and T-NC would negatively affect both optical & electrical gain, so ensuring the perfect contact between layers is of outermost importance.
We have added a section in lines 286-294 speaking about the trade-off between the optical & electrical gains and highlighting the importance of tight adjoining between the active layer material and the T-NC Al nanostructure:
“It should be noted that the implementation of tight adjoining between the active layer material and the T-NC Al nanostructure will strongly influence both the optical gain and the electrical gain of the solar cell. Roughness induced by the nanostructured Al electrode would deteriorate the contact characteristics of the active layer and the Al electrode, which might worsen the transport and collection of charge carriers at the contact interface and thus weaken the performance of solar cells [5–7]. However, in case of perfect adjoining between the active layer material and the T-NC Al nanostructure, the latter would facilitate charge carriers’ collection and improve the short-circuit current density (JSC) [5].”

Reviewer 3 Report (New Reviewer)
In lines 41-42 the Authors pointed on a need to increase absorption of sunlight photons in thin layer solar cells. However, at mentioning various methods applied in this direction they have avoided most prominent, cheap and easy to apply method of solar cell meallization utilizing the giant plasmonic effect in nano-scale – this has been demonstrated both experimentally and theoretically. It has been shown that the increase of the photoresponse of photovoltaic Si diode setup can reach even 200%, mostly due to increase of the photon absorption mediated by localized plasmons (Appl. Phys. Lett. 86 (2005) 063106 and many other papers cited e.g., in monograph “Quantum Nano-Plasmonics” Cambridge UP (2020)).
Tsuchime-like nano-crater aluminum (T-NC Al) film – this term should more precisely explaned (as it is not in common use).
The observation of the Authors “The effect of using a T-NC film is that the higher, the thinner the active layer of the solar panel” strongly supports the plasmomic effect, not only enhancement of photon path length. Conclusion in 63-65 lines is also simplified. If actually the mentioned dependence of absorption increase versus wave-length takes place, it arises a question what is an origin of such a behavior. Again the Authors did not here take into account plasmonic effect in metallic nanostructure and only rely on numerical simulation of rather geometry effect via conventional Comsol system.
The comparison of various substrates should include perovskite substrate, as the plasmonic effect in perovskite solar cell is different than in p-n junction cells like Si cells (Nano Energy 75 (2020) 104751) – in particular, the efficiency enhancement due to metallization is not related with absorption increase in perovskite, but with decreasing of binding energy of excitons. Such a comparison (including perovskite thin film cell) would allow an identification of a true mechanism (which is probably complex and involves different factors and different mechanisms at the same time).
In caption of Fig. 1 the wording addressed to f) panel is unclear.
It has been shown that Comsol gives 90% error in assessment of light absorption in metalized solar cells (Nanomaterials 9 (1) (2018) 3). Even if authors do not include indicated in the literature corrections allowing the reduction of the discrepancy between Comsol simulations and experiments, they should at least comment on their awareness of Comsol limitations (otherwise, their results are not trustworthy). Comsol in the version used by the Authors is unable to account for strong absorption icrease due to direct near-field interaction of plasmons in nano caves of considered Al structure with substrate seconductor band electrons. Comsol can only solve Maxwell equations for a Fresnel type boundary problem and can correctly desribe local concentration of electric field of incident e-m wave of sunlight photons – this, however, gives only 5-10% of absorption increase compared to ca. 90% of experimentally observed (Nanomaterials 9 (1) (2018) 3). The Authors should address to these already known in literature studies which may be of significance for their simulation.
The physical model used by the Authors is highly simplified – it does not take into account the nano-optical effects in the microscale of multi-layer structure including nano-corrugated metal layer. The model in the submission is limited to rather geometric boundary effect (Fresnel-Maxwell problem) and this must be clearly stated claryfying which effects are included and which are not. The submission would serve rather as the demonstration of some simplified approach (numerical simulation upon conventional Comsol) and any conclusions must be associated by comments which effects were not included. Moreover, in the discussion of the results, the authors listed only the observations from the simulation, without any explanation of the observed trends within the physical mechanism.
This causes a need to revise the submission by increasing an insight into physical effects in the considered structure. Otherwise, the conclusions may not be trustworthy. In addition, the authors stopped at simulating Comsol only without confronting any experiment.
Summarising, the paper must be revised in the layer of physical mechanism responsible for absorption increase due to application of metallic layer structuralized in nano scale. Helpful, would be mentioned ebove references (in addition also a remarkable experimental result might be of significance here, Appl. Phys. Lett. 69 (1996) 2327).
some small linguistic errors must be corrected -- medium level proofreading is of order
Author Response
We would like to thank Reviewer 3 for evaluating our work and providing insightful comments. We will address the comments section by section.
In lines 41-42 the Authors pointed on a need to increase absorption of sunlight photons in thin layer solar cells. However, at mentioning various methods applied in this direction they have avoided most prominent, cheap and easy to apply method of solar cell meallization utilizing the giant plasmonic effect in nano-scale – this has been demonstrated both experimentally and theoretically. It has been shown that the increase of the photoresponse of photovoltaic Si diode setup can reach even 200%, mostly due to increase of the photon absorption mediated by localized plasmons (Appl. Phys. Lett. 86 (2005) 063106 and many other papers cited e.g., in monograph “Quantum Nano-Plasmonics” Cambridge UP (2020)).
We are grateful for the valuable comment. Indeed, utilizing the giant plasmonic effect in nano-scale is a viable solution towards increasing absorption of sunlight. We decided to add this method to lines 43-46 as suggested by Reviewer III:
“Another approach relies on increasing the photon absorption mediated by local plasmons, which is cheap and relatively easy to apply, but restricted to particular materials and affecting a limited wavelength range [8,9].”
However, the method of increasing the photon absorption mediated by localized plasmons is restricted to particular materials and limited in the affected wavelength range. For instance, T-NC nanostructure can be obtained through nanoporous electrochemical anodizing and leaves such popular plasmonic materials such as gold (Au) and silver (Ag) out of question. This method relies on the inherent property of the valve metals, such as aluminum (Al), titanium (Ti), tantalum (Ta), niobium (Nb), tungsten (W), chromium (Cr) and other. Among the valve metals, Al attracts more interest for optoelectronic applications due to excellent conductivity and possibility of forming highly transparent conductive films without lithography [1,10]. However, Al is less attractive than Ag and Au for plasmonic applications since the localized surface plasmons in Al affect the short wavelength (UV) region [11–13] – below the range studied in the paper.
Tsuchime-like nano-crater aluminum (T-NC Al) film – this term should more precisely explaned (as it is not in common use).
The pattern resulting from the electrochemical anodizing of Al followed by etching the alumina (Al2O3) closely resembles the so-called Tsuchime pattern resulting from hammering the metal – yet at nanoscale. We mention the resemblance in lines 94-96 in the article:
“As can be seen from the figure, the nanopatterned structure resembles a crater left by a hammer on a metallic surface – the so-called hammertone or Tsuchime technique [14].”
The observation of the Authors “The effect of using a T-NC film is that the higher, the thinner the active layer of the solar panel” strongly supports the plasmomic effect, not only enhancement of photon path length. Conclusion in 63-65 lines is also simplified. If actually the mentioned dependence of absorption increase versus wave-length takes place, it arises a question what is an origin of such a behavior. Again the Authors did not here take into account plasmonic effect in metallic nanostructure and only rely on numerical simulation of rather geometry effect via conventional Comsol system.
We thank the Reviewer III for this comment. Actually, the Figure 5 demonstrates that the effect of T-NC film is that the higher, the thinner the active layer of the solar panel. The figure shows varying the active layer thickness – not the size of the nanopatterns. We could have agreed with the Reviewer that it is related to plasmonic effect if it (a) relied on the dimensions of NCs, and (b) if the material of T-NC in question was Au or Ag. In fact, the effect of T-NC is more pronounced with thinner active layers because of pure ray optical effect – more range of wavelength reach T-NC through the thinner active layer and get reflected by T-NC compared to thicker active layer. In thicker active layers, longer wavelength are more affected – which is evident from the Figure 5d. In the Figure 5d, the maximum absorbance increase when HSi equals to 500 nm, 1, 3, and 10 µm, happens at wavelength λ = 620, 680, 850, and 950 nm, respectively. So it “moves” to longer wavelength (to NIR) when increasing the active layer thickness without changing any parameters of T-NC nanostructure. We believe that this is clearly ray optical effect and conventional numerical simulation via Comsol system is sufficient and it is not necessary here take into account plasmonic effect in aluminium T-NC.
The comparison of various substrates should include perovskite substrate, as the plasmonic effect in perovskite solar cell is different than in p-n junction cells like Si cells (Nano Energy 75 (2020) 104751) – in particular, the efficiency enhancement due to metallization is not related with absorption increase in perovskite, but with decreasing of binding energy of excitons. Such a comparison (including perovskite thin film cell) would allow an identification of a true mechanism (which is probably complex and involves different factors and different mechanisms at the same time).
We thank the Reviewer III for the valuable insight and for citing a prominent report on metallization of solar cells, exciton channel of plasmon photovoltaic effect in perovskite cells. In the abovementioned article, the application of dilute surface coverings of cells with Au, Ag or Cu nanoparticles of size is being studied. Indeed, strong resonance of surface plasmons in metallic nanoparticles is responsible for the effects mentioned in the paper, however, in the current manuscript T-NC structure is based on Al, which has pronounced plasmonic effect below 400 nm (out of range of the study). Anyway, we added an information on the mentioned effects in perovskite solar cell in lines 262-268 for consideration as a second channel of improving the efficiency of perovskite solar cells not relying solely onto absorption enhancement:
“It is important to mention that sole enhancement of the light absorption is not the only mechanism to increase the efficiency of perovskite solar cells where the large increase of the final efficiency of a cell has been observed due to the plasmon effect related with the influence of metallic components onto internal electricity of a cell [15–17]. However, the current study focuses on Al nanostructures, where the localized surface plasmon resonance (LSPR) affects the wavelength region below 400 nm [11,12].”
In caption of Fig. 1 the wording addressed to f) panel is unclear.
We thank the Reviewer III for pointing our attention to that matter. Figure 1f is showing a scanning electron microscope (SEM) of T-NC nanostructure fabricated by our group. We have added a scale bar information to the figure caption.
It has been shown that Comsol gives 90% error in assessment of light absorption in metalized solar cells (Nanomaterials 9 (1) (2018) 3). Even if authors do not include indicated in the literature corrections allowing the reduction of the discrepancy between Comsol simulations and experiments, they should at least comment on their awareness of Comsol limitations (otherwise, their results are not trustworthy). Comsol in the version used by the Authors is unable to account for strong absorption icrease due to direct near-field interaction of plasmons in nano caves of considered Al structure with substrate seconductor band electrons. Comsol can only solve Maxwell equations for a Fresnel type boundary problem and can correctly desribe local concentration of electric field of incident e-m wave of sunlight photons – this, however, gives only 5-10% of absorption increase compared to ca. 90% of experimentally observed (Nanomaterials 9 (1) (2018) 3). The Authors should address to these already known in literature studies which may be of significance for their simulation.
The physical model used by the Authors is highly simplified – it does not take into account the nano-optical effects in the microscale of multi-layer structure including nano-corrugated metal layer. The model in the submission is limited to rather geometric boundary effect (Fresnel-Maxwell problem) and this must be clearly stated claryfying which effects are included and which are not. The submission would serve rather as the demonstration of some simplified approach (numerical simulation upon conventional Comsol) and any conclusions must be associated by comments which effects were not included. Moreover, in the discussion of the results, the authors listed only the observations from the simulation, without any explanation of the observed trends within the physical mechanism.
We would like to thank the Reviewer III for the valuable comment on the limitations of the theoretical modelling of metalized solar cells. We understand the Reviewer’s concerns, but, again, we would have strongly agreed if the metallization of interest was Au or Ag metallization as discussed in Nanomaterials 9 (1) (2018) 3 or Nano Energy 75 (2020) 104751 mentioned above. However, the current study focuses on the Al nanostructure, where the related effects are present in the ultra-violet (UV) region below investigated in the paper, with the exception of slight effect of interband electron transition in Al [11,18]. What we observe in the current study is related to the ray optics where COMSOL FEM performs with high efficiency and the accuracy of the method is very high. We have further added a notice that the COMSOL simulator has limitations in cases where the nanostructure under consideration for a solar cell falls within the range of influence of plasmonic effects for a given nanostructure material in lines 115-118 in the manuscript:
“It should be noted that the COMSOL simulator has limitations in cases where the nanostructure under consideration for a solar cell falls within the range of influence of plasmonic effects for a given nanostructure material [19].”
some small linguistic errors must be corrected -- medium level proofreading is of order
We want to thank the Reviewer III for pointing our attention to that matter. We have used a proofreading service for our manuscript.

Round 2
Reviewer 1 Report (New Reviewer)
The authors have made a great effort answering, in my opinion, quite convincingly, the different questions raised by the reviewers. I consider that it can be published.
Author Response
We want to thank Reviewer I for analysing our study and providing comments and suggestions for improving the manuscript.
Reviewer 2 Report (New Reviewer)
The authors properly answer all the questions from the reviewers. The manuscript can be published.
Author Response
We want to thank Reviewer II for analysing our study and providing comments and suggestions for improving the manuscript.
Reviewer 3 Report (New Reviewer)
The Authors have introduced corrections. Generaly they have improved the submission. Though the details of physical mechanism related to efficiency increase deserve further study, some important aspects have been commented - in my opinion the inresting direction would be an analysis of the absorption gain with respect to an average size of craters in Al layer (these caves are dual to naparticles from point of view of plasmon resonances and should support a similar size effect as for nanoparticles), this would allow to distinguish the geometrical effect of light scattering (and increase of photon path) from plasmon coupling channel. Maybe in conclusion similar comment would be of order.
final linguistic proofreading is suggeste
Author Response
We would like to thank Reviewer III for thoroughly analysing our study and providing useful comments and suggestions which we found helpful for improving the manuscript. We believe that considering that (a) the plasmon resonance in Al appears in the shorter (UV) wavelength region <400 nm and (b) the thickness of the active layers’ measures at least several hundred nm, it will mask the possible influence of the plasmon resonance on the structure under study. However, on the other hand, if we push the plasmon resonance effect towards longer wavelengths, say, by applying a thin (5-10 nm) layer of another metal (Au or Ag) on top of the T-NC nanostructure, we would be able to benefit from strongly increased absorption through plasmon resonance. In fact, it gave us the idea that it would be interesting to investigate a modified structure including an ultra-thin (5-10 nm) coating of another metal (such as Au or Ag) – to trigger a plasmonic effect, which will surely boost the absorption stronger than by 80.3% (maximum achieved with bare T-NC).
Considering the above-mentioned, we have added this observation in the Discussion (lines 264-267):
“Covering the T-NC nanostructure with an ultra-thin (5-10 nm) layer of another metal, such as gold (Au) or silver (Ag), onto the T-NC nanostructure can shift the plasmonic effect from the UV towards the visible wavelength range [1].”
and in the Conclusion (lines 313-315):
“A further direction of research may include employing a plasmonic effect, for example, by adding an ultra-thin layer of another metal such as gold or silver onto the T-NC nanostructure.”
This manuscript is a resubmission of an earlier submission. The following is a list of the peer review reports and author responses from that submission.
Round 1
Reviewer 1 Report
This manuscript by Marus et al. demonstrated a tsuchime-like aluminum film by electrochemical anodization for improving the absorbance for the silicon solar cells. This nanostructure can be easily formed and cost-effective without expensive process like lithography. The authors also showed the theoretical studies on the effects of diameter of nano-craters and the thickness of the active silicon layer. The discussion and conclusion based on the results are reasonable and well organized. However, I’m not sure this manuscript has novelty enough to publish in this Nanomaterials. Authors fabricated this structure and showed in SEM images, however, didn’t applied to the solar cells. It would be better if this nanostructure is tested to the real silicon cells to prove how this structure works well as theoretical estimation.
Reviewer 2 Report
The authors have shown that using textured Al back contacts in thin and ultra-thin silicon photovoltaic absorbers can increase light absorption. However, the structure they study assumes that the silicon is directly grown on the textured Al, which seems impractical. Can the authors please address in the paper how introducing an air gap between the flat single crystal silicon surface and the textured Al would affect their results?
Minor comments:
Page 2 line 57: "The effect of using a T-NC film is that the higher, the thinner the active layer of the solar panel." The meaning is unclear.
Page 3 line 111: "non-casual" should be "non-causal"